# MAPE-PPI: Towards Effective and Efficient Protein-Protein Interaction Prediction via Microenvironment-Aware Protein Embedding

**Lirong Wu**[1,2], **Yijun Tian**[3], **Yufei Huang**[1], **Siyuan Li**[1], **Haitao Lin**[1], **Nitesh V Chawla**[3], **Stan Z. Li**[1,†]
[1]Westlake University, [2]Zhejiang University, [3]University of Notre Dame
`{wulirong, huangyufei, lisiyuan, linhaitao stan.zq.li}@westlake.edu.cn`
`{yijun.tian, nchawla}@nd.edu`

## Abstract

Protein-Protein Interactions (PPIs) are fundamental in various biological processes and play a key role in life activities. The growing demand and cost of experimental PPI assays require computational methods for efficient PPI prediction. While existing methods rely heavily on protein sequence for PPI prediction, it is the protein structure that is the key to determine the interactions. To take both protein modalities into account, we define the microenvironment of an amino acid residue by its sequence and structural contexts, which describe the surrounding chemical properties and geometric features. In addition, microenvironments defined in previous work are largely based on experimentally assayed physico-chemical properties, for which the "vocabulary" is usually extremely small. This makes it difficult to cover the diversity and complexity of microenvironments. In this paper, we propose *Microenvironment-Aware Protein Embedding for PPI prediction* (MPAE-PPI), which encodes microenvironments into chemically meaningful discrete codes via a sufficiently large microenvironment "vocabulary" (i.e., codebook). Moreover, we propose a novel pre-training strategy, namely *Masked Codebook Modeling* (MCM), to capture the dependencies between different microenvironments by randomly masking the codebook and reconstructing the input. With the learned microenvironment codebook, we can reuse it as an off-the-shelf tool to efficiently and effectively encode proteins of different sizes and functions for large-scale PPI prediction. Extensive experiments show that MAPE-PPI can scale to PPI prediction with millions of PPIs with superior trade-offs between effectiveness and computational efficiency than the state-of-the-art competitors. Codes are available at: https://github.com/LirongWu/MAPE-PPI.

## 1 Introduction

Protein-Protein Interactions (PPIs) (Lv et al., 2021; Hu et al., 2021; Wu et al., 2022a) perform specific biological functions that are essential for all living organisms. Therefore, understanding, identifying, and predicting PPIs are critical for medical, pharmaceutical, and genetic research. Over the past decades, a number of experimental methods (Shoemaker & Panchenko, 2007; Zhou et al., 2016) have been proposed for high-throughput PPI prediction, such as Yeast Two-Hybrid Screens (Fields & Song, 1989) and Mass Spectrometric Protein Complex Identification (Ho et al., 2002), etc. However, experimental PPI assays in the wet laboratory are usually very expensive and time-consuming, making it hard to match the growing demand for large-scale PPI prediction, which calls for developing computational methods to predict unknown PPIs more **efficiently** and **effectively**.

With the development of artificial intelligence techniques, computational methods for PPI prediction have undergone a paradigm shift from machine learning techniques (Sarkar & Saha, 2019; Liu et al., 2018; Zhang et al., 2017) to deep learning techniques (Hashemifar et al., 2018; Zhao et al., 2023). Existing deep learning-based methods can be mainly divided into two categories: sequence-based methods and structure-based methods. The sequence-based methods (Zhao et al., 2023; Wang et al., 2019; Zhang et al., 2019a) extract protein representations from primary amino acid sequences by Convolutional Neural Networks (CNNs) (LeCun et al., 1995), Recurrent Neural Networks (RNN) (Armenteros et al., 2020) or Transformer (Vaswani et al., 2017), and then directly take the repre-

sentations of two proteins to predict their interactions. However, it is the structure of a protein, not the sequence, that determines the protein's function and interactions with others. Therefore, structure-based methods (Gao et al., 2023b; Yuan et al., 2022; Bryant et al., 2022) typically use Graph Neural Networks (GNNs) to model protein 2D or 3D structures and formulate protein representation learning as well as PPI prediction in a unified framework. While structure-based methods usually outperform sequence-based methods in terms of prediction accuracy, they may suffer from a heavy computational burden and are only applicable to small-scale PPI prediction with 10,000 PPIs.

There are twenty types of amino acid residues in nature, but the distribution of different residues is not uniform, and more importantly, the residues of the same type may exhibit different physicochemical properties due to different microenvironments. The microenvironments of different residues may also be similar or overlap. If the microenvironments of two residues are highly identical, it would be redundant to encode them separately at each training epoch. If we can learn a "vocabulary" (i.e., codebook) that records those most common microenvironmental patterns, we can use it as an off-the-shelf tool to encode proteins into microenvironment-aware embeddings. Thus, training efficiency can be improved by decoupling end-to-end PPI prediction into two stages of learning the microenvironment codebook and codebook-based protein encoding. There have been some previous works that define the sequence and structural contexts surrounding a target residue as its microenvironment and classify it into a dozen types based on its physicochemical properties (Huang et al., 2023a; Lu et al., 2022), which constitutes a special microenvironment "vocabulary". However, unlike the rich word vocabulary in languages, the experimental microenvironment "vocabulary" is usually coarse-grained and limited in size. Therefore, to learn a fine-grained vocabulary with rich information, we propose a variant of VQ-VAE (Van Den Oord et al., 2017) for microenvironment-aware protein embedding, which consists of two aspects of designs: *(1) Microenvironment Discovery:* learning a large and fine-grained microenvironment vocabulary (i.e., codebook); and (2) *Microenvironment Embedding:* encoding microenvironments into chemically meaningful discrete codes using the codebook. Furthermore, we propose a novel codebook pre-training task, namely *Masked Codebook Modeling* (MCM), to capture the dependencies between different microenvironments by randomly masking the codebook instead of input features or discrete codes. With the learned microenvironment codebook, we can take it as an off-the-shelf tool to encode proteins into microenvironment-aware protein embeddings. The obtained embeddings can be further utilized as node features on a large-scale PPI graph network, which is then encoded by GNNs to predict PPIs.

Considering efficiency and effectiveness as two important criteria, we report the micro-F1 scores and the training time for different algorithms on the SHS27k dataset that contains 16,912 PPIs in Fig. 1. It can be seen that those sequence-based methods are efficient in training time, but come with suboptimal performance, due to the lack of modeling critical structural information. In contrast, structure-based methods, such as HIGH-PPI (Gao et al., 2023b), can achieve fairly satisfactory performance by benefiting from protein structures, but the computational burden is overly high. Specifically, HIGH-PPI creates a hierarchical graph in which a node in the PPI net-

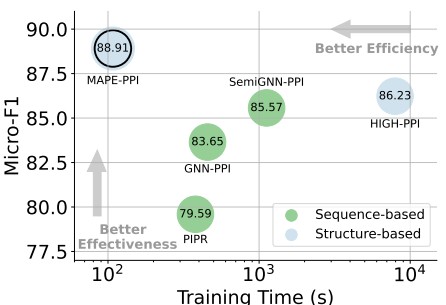

Figure 1: Micro-F1 *vs.* Training Time.

work is a protein graph, and each node in the protein graph is a residue. Despite the HIGH-PPI's advantages in terms of prediction performance, it may take more than 200 hours of training time to model a large STRING dataset with one million PPIs, making it hard to scale up and heavily depend on computation resources. As a comparison, our proposed MPAE-PPI inherits the high computational efficiency of sequence-based methods, while enjoying the structure awareness of structure-based methods, showing the supriority trade-offs between efficiency and effectiveness.

The contributions of this paper are summarized as (1) To encode both the protein sequence and structure effectively, we define the microenvironment of a residue based on its sequential and structural contexts. (2) We propose a variant of VQ-VAE to discover fine-grained microenvironments and learn microenvironment-aware protein embeddings. (3) We propose *Masked Codebook Modeling* that directly masks the codebook instead of input features or discrete codes to capture the dependencies between different microenvironments. (4) We scale MPAE-PPI to large-scale PPI prediction with millions of PPIs with better effectiveness and efficiency than the state-of-the-art competitors.

## 2 RELATED WORK

**Protein-Protein Interaction (PPI).** The identification of PPIs plays a very important role in drug discovery. However, PPI prediction by various experimental assays (Fields & Song, 1989; Ho et al., 2002) is very expensive and time-consuming, and it is hard to generalize to unknown PPIs. The prediction of PPIs by protein docking (Zhang et al., 2016; Tsuchiya et al., 2022) or molecular dynamics simulations (Tan et al., 2015; Rakers et al., 2015; Kumar & Yaduvanshi, 2023) has been studied for decades with great success. However, these simulation-based methods may not work well in some cases, such as when the exact 3D protein structure is unknown, and often suffer from computational bottlenecks. Recent advances in deep learning (DL) provide new insights into reducing the reliance on exact 3D protein structures and developing DL-based methods for more efficient PPI prediction.

Different from earlier machine learning-based methods (Chen et al., 2019; Wong et al., 2015) that directly map pairs of protein sequence features to predict interactions, existing DL-based methods mainly focus on two aspects: (1) learning protein representations, and (2) inferring potential PPIs. The latter aspect has been well studied in the past few years, and the prevailing scheme nowadays is to first construct a graph-structured PPI network from known PPIs (Yang et al., 2020), and then predict those unknown interactions from the graph topology by using GNNs (Lv et al., 2021; Zhao et al., 2023). Protein representation learning aims to learn meaningful and informative embeddings from proteins, and the existing methods can be divided into two main categories: sequence-based and structure-based. The sequence-based methods (Wang et al., 2019; Zhang et al., 2019a) generally extract representations from protein sequences by CNN, RNN, or Transformer, the structure-based methods use GNNs to model protein structural information at the residue level and finally obtain graph-level protein representations by global pooling (Gao et al., 2023b). Previous work has mostly focused on the effectiveness of PPI prediction (e.g., prediction accuracy), but comparatively little work explored the training efficiency, which is very important for large-scale PPI analysis.

**Pre-training on Proteins.** Tremendous efforts have been devoted to protein pre-training (Wu et al., 2024; 2022a; Gao et al., 2023a; Lin et al., 2023; Huang et al., 2023b;c), whose main goal is to extract transferable knowledge from massive unlabeled data and then generalize it to protein applications. Early work analogizes an amino acid in a protein sequence to a word in a sentence and directly extends pre-training tasks for natural language processing to protein sequences, including Masked Language Modeling (MLM) (Rives et al., 2021; Elnaggar et al., 2020), Contrastive Predictive Coding (CPC) (Lu et al., 2020), and Next Amino acid Prediction (NAP) (Alley et al., 2019), etc. While sequence-based pre-training has been shown to be effective, structure-based pre-training is a better solution since the function of a protein is mainly determined by its structure. Typical methods for structure pre-training include Structure Contrastive Learning (Zhang et al., 2022), Distance/Angle Prediction (Chen et al., 2022), and Maksed Structure Modeling (MSM) (Yang et al., 2022), etc.

**Masked Modeling** is one of the most popular and successful means for pre-training on proteins, applicable to both sequence and structure modeling. First proposed by BERT (Devlin et al., 2018) for natural language processing, masked modeling has later been extended to computer vision and graphs, yielding classical works, such as MAE (He et al., 2022) and GraphMAE (Hou et al., 2022). Following this line in this paper, but unlike previous works that mask input features or hidden codes (Peng et al., 2022) for *training a better image encoder or decoder*, we randomly mask the microenvironment codes in the codebook to capture their dependencies for *learning a better codebook*.

## 3 PRELIMINARY

### 3.1 PROBLEM STATEMENT

Given $N$ proteins $\mathcal{P} = \{P_1, P_2, \cdots, P_N\}$ and a set of PPIs $\mathcal{X} = \{(P_i, P_j) \mid P_i, P_j \in \mathcal{P}, i \neq j\}$, we can construct a PPI graph $\mathcal{G}_T = (\mathcal{V}_T, \mathcal{E}_T)$, where each node $p_i \in \mathcal{V}_T$ represents a protein $P_i \in \mathcal{P}$ and each edge $e_{i,j} \in \mathcal{E}_T$ iff $(P_i, P_j) \in \mathcal{X}$. Consider a semi-supervised PPI prediction task in which only a subset of PPIs $\mathcal{X}_L \in \mathcal{X}$ with corresponding interaction types $\mathcal{Y}_L$ are known, we can denote the labeled set as $\mathcal{D}_L = (\mathcal{X}_L, \mathcal{Y}_L)$ and unlabeled set as $\mathcal{D}_U = (\mathcal{X}_U, \mathcal{Y}_U)$, where $\mathcal{X}_U = \mathcal{X} \backslash \mathcal{X}_L$. The objective of PPI prediction aims to learn a mapping $\mathcal{F} : \mathcal{X} \rightarrow \mathcal{Y}$ on the training data $(\mathcal{X}, \mathcal{Y}_L)$ in the transductive setting, so that it can be used to infer the interaction types $\mathcal{Y}_U$ of unlabeled PPIs $\mathcal{X}_U$.

### 3.2 HETEROGENEOUS PROTEIN GRAPH CONSTRUCTION

A protein $P_i \in \mathcal{P}$ with $M$ amino acid residues can be denoted by a string of its primary sequence, $S_i = (v_1, v_2, \cdots, v_M)$, where each residue $v_m \in S_i$ is one of the 20 amino acid types. The

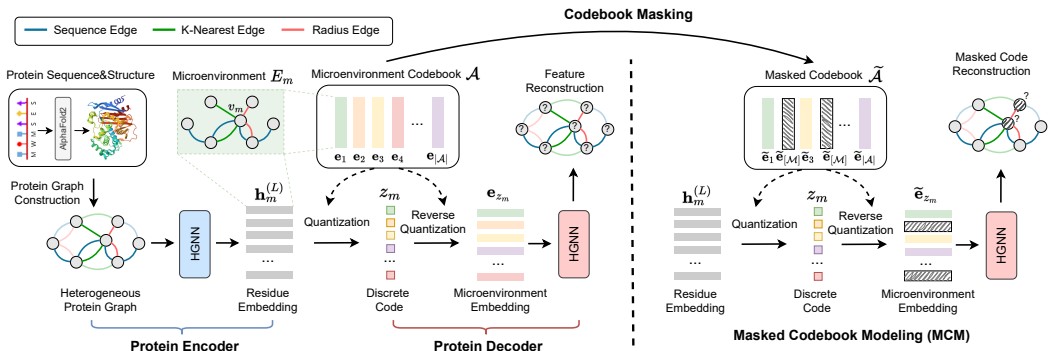

Figure 2: Left: Illustration of microenvironment discovery and microenvironment-aware protein embedding. Right: Illustration of pre-training the codebook by Masked Codebook Modeling (MCM).

amino acid sequences of proteins can be folded into stable 3D structures in the real physicochemical world. Therefore, we jointly represent the sequence and structure of protein $P_i$ as a residue-level heterogeneous protein graph $\mathcal{G}_P^{(i)} = (\mathcal{V}_P^{(i)}, \mathcal{E}_P^{(i)}, \mathcal{R})$, where $\mathcal{V}_P^{(i)}$ is the node set of $M$ residues, and $\mathcal{E}_P^{(i)}$ is the set of edges that connects the residues (Zhang et al., 2022). There is a total of three different sequential and structural edge types $\mathcal{R}$ adopted in this paper, including *(i)* sequential edge: two residues connected in sequence; *(ii)* radius edge: two residues with a spatial Euclidean distance less than a threshold $d_r$; and *(iii)* $K$-nearest edge: two residues that are spatial $K$-nearest neighbors.

### 3.3 DEFINITION OF RESIDUE-LEVEL MICROENVIRONMENT

The microenvironment of a residue describes its surrounding chemical and geometric features based on the sequential and structural contexts. Formally, we define the microenvironment $E_m$ of residue $v_m$ as a $v_m$-ego subgraph of the protein graph $\mathcal{G}_P^{(i)}$, with its node set $V_{E_m} \subseteq V_P^{(i)}$ defined as

$$V_{E_m} = \left\{ v_n \mid |m - n| \le d_s, \|C\alpha_m - C\alpha_n\| \le d_r, v_n \in \mathcal{N}_m^{(K)} \right\}, \tag{1}$$

where $d_s$ and $d_r$ are cut-off distances, $C\alpha_m$ and $C\alpha_n$ are the 3D coordinates of carbon-alpha atoms in residues $v_m$ and $v_n$, and $\mathcal{N}_m^{(K)}$ is the $K$-hop neighborhood of residue $v_m$ in the 3D space. In this paper, we aim to learn both a microenvironment codebook $\mathcal{A}$ and a mapping $\mathcal{T} : E_m \to \mathcal{A}$ to achieve microenvironment discovery and protein embedding in an end-to-end manner.

## 4 MICROENVIRONMENT-AWARE PROTEIN EMBEDDING

### 4.1 HETEROGENEOUS PROTEIN ENCODING

Graph Neural Networks (GNNs) (Kipf & Welling, 2016; Wu et al., 2021; 2022c;b; 2023) have demonstrated their superior capability in handling graph-structured data. Given a heterogeneous protein graph $\mathcal{G}_P = (\mathcal{V}_P, \mathcal{E}_P)$ with different edge types $\mathcal{R}$, we can apply a Heterogeneous Graph Neural Networks (HGNN) (Zhang et al., 2019b) as the protein encoder, where each edge type $r \in \mathcal{R}$ corresponds to a convolutional kernel $\boldsymbol{W}_r$. The heterogeneous protein encoder consists of three key computations for each node $v_m$ at every encoder layer: (1) aggregating messages from the neighborhood $\mathcal{N}_r(m) = \{v_n | e_{m,n} = r\}$ of node $v_m$ that corresponds to the edge type $r$ ($r \in \mathcal{R}$); (2) applying a linear transformation $\boldsymbol{W}_r$ for each edge type $r$; and (3) summing updated messages from the neighborhoods of different edge types by applying a linear transformation $\boldsymbol{W}_h$. Considering a $L$-layer heterogeneous protein encoder, the formulation of the $l$-th layer can be defined as follows:

$$\mathbf{h}_m^{(l)} = \mathrm{BN}\left(\mathrm{ReLU}\left(\boldsymbol{W}_h^{(l)} \cdot \sum_{r \in \mathcal{R}} \boldsymbol{W}_r^{(l)} \sum_{v_n \in \mathcal{N}_r(m)} \mathbf{h}_n^{(l-1)}\right)\right), \text{for } 1 \le l \le L, \tag{2}$$

where $\mathbf{h}_m^{(0)} = \mathbf{x}_m$ is the input feature of residue $v_m$, $\mathbf{h}_m^{(l)}$ is the node embedding of residue $v_m$ in the $l$-th layer, $\mathrm{BN}(\cdot)$ denotes a batch normalization layer, and $\mathrm{ReLU}(\cdot)$ is the activation function.

### 4.2 MICROENVIRONMENT CODEBOOK DISCOVERY AND EMBEDDING

The heterogeneous protein encoder models the microenvironment of a residue by aggregating messages from different neighborhoods, i.e., sequential and structural contexts. The obtained embedding

$\mathbf{h}_m^{(L)}$ contains all the information about its microenvironment $E_m$. Given a microenvironment codebook $\mathcal{A} = \{\mathbf{e}_1, \mathbf{e}_2, \cdots, \mathbf{e}_{|\mathcal{A}|}\} \in \mathbb{R}^{|\mathcal{A}| \times F}$, the microenvironments $\{E_1, E_2, \cdots, E_M\}$ of a protein can be tokenized to discrete codes $\{z_1, z_2, \cdots, z_M\}$ by vector quantization (Van Den Oord et al., 2017) that looks up the nearest neighbor in the codebook $\mathcal{A}$ for each embedding $\mathbf{h}_m^{(L)}$:

$$z_m = \operatorname{argmin}_n \left\| \mathbf{h}_m^{(L)} - \mathbf{e}_n \right\|_2, \quad \text{where } 1 \leq m \leq M. \tag{3}$$

Next, we focus on how to construct and learn a suitable microenvironment codebook $\mathcal{A}$, i.e., microenvironment discovery. A natural idea is to heuristically construct the codebook $\mathcal{A}$ using categorization of experimentally assayed microenvironments (Fields & Song, 1989), but the size of such a codebook with only a dozen available types may be too small and coarse-grained due to the high cost of experimental assays. To tackle this problem, we directly parameterize the codebook $\mathcal{A}$ as learnable variables, and then follow the VQ-VAE (Van Den Oord et al., 2017) to take data reconstruction as a pretext task to simultaneously learn the codebook $\mathcal{A}$ and train the heterogeneous protein encoder. Specifically, we use a symmetric variant of heterogeneous protein encoders as the protein decoder, which outputs the reconstructions $\{\widehat{\mathbf{x}}_m\}_{i=1}^M$ of the original residue features $\{\mathbf{x}_m\}_{i=1}^M$ from the discrete codes $\{z_m\}_{m=1}^M$. The training objective for protein $P_i \in \mathcal{P}$ is defined as follows:

$$\mathcal{L}_{\text{VQ}} = \frac{1}{M} \sum_{m=1}^M \left( \left\| \mathbf{x}_m - \widehat{\mathbf{x}}_m \right\|_2^2 + \left\| \operatorname{sg}\big[\mathbf{h}_m^{(L)}\big] - \mathbf{e}_{z_m} \right\|_2^2 + \beta \left\| \mathbf{h}_m^{(L)} - \operatorname{sg}\big[\mathbf{e}_{z_m}\big] \right\|_2^2 \right), \tag{4}$$

where $\beta$ is a trade-off hyper-parameter, and the stop-gradient operation $\operatorname{sg}[\cdot]$ is used to resolve the non-differentiability of the vector quantization by straight-through estimator (Bengio et al., 2013). The roles of the three losses defined in Eq. (4) are as follow: (1) the first term is a reconstruction loss, used to train both encoder and decoder; (2) the second term is a codebook loss, used to update the codebook to make the microenvironment vectors close to the most similar embeddings; (3) the third term is a commitment loss, used to encourage the output of the encoder to stay close to the chosen codebook vector by only training the encoder. In such a way, we can unify the microenvironment discovery, codebook updating, and encoder training, in a unified loss function of Eq. (4).

### 4.3 MASKED CODEBOOK MODELING (MCM)

Masked modeling has been widely used in various deep learning applications, such as masked language modeling, masked image modeling, and graph masking autoencoder. The core idea of these works is to randomly mask input features and then reconstruct the inputs from the masked data with the purpose of training a more powerful and context-aware encoder. However, recent work has found that random masking can lead to meaningless semantics, resulting in distribution bias between training and test data (Li et al., 2022; Shi et al., 2022). For example, AttMask (Kakogeorgiou et al., 2022) has studied the problem of which tokens to mask and proposes an attention-guided masking strategy to make informed decisions. In this paper, we propose to directly mask the codebook $\mathcal{A} = \{\mathbf{e}_1, \mathbf{e}_2, \cdots, \mathbf{e}_{|\mathcal{A}|}\}$ with explicit semantics instead of input features $\{\mathbf{x}_m\}_{m=1}^M$ or hidden codes $\{z_m\}_{m=1}^M$. The motivation behind this is that there are generally overlaps (dependencies) in the microenvironments of two residues that are adjacent in sequence or structure. To capture such microenvironmental dependencies in the codebook, we propose Masked Codebook Modeling (MCM) to mask the codebook $\mathcal{A}$ and then reconstruct the inputs based on the unmasked codes, *aiming to learn a more expressive codebook $\mathcal{A}$ rather than a more powerful protein encoder*. Formally, we sample a subset of codes $\mathcal{M} \subset \mathcal{A}$ and mask them with a mask vector [MASK] with $\mathbf{e}_{[\mathcal{M}]} \in \mathbb{R}^F$. Therefore, the microenvironment vector $\widetilde{\mathbf{e}}_m$ in the masked codebook $\widetilde{\mathcal{A}}$ is defined as

$$\widetilde{\mathbf{e}}_m = \begin{cases} \mathbf{e}_{[\mathcal{M}]} & \mathbf{e}_m \in \mathcal{M} \\ \mathbf{e}_m & \mathbf{e}_m \notin \mathcal{M} \end{cases}. \quad \text{where } 1 \leq m \leq M \tag{5}$$

Next, we map each discrete code $z_m$ back to the corresponding embedding $\widetilde{\mathbf{e}}_{z_m}$ based on the masked codebook $\widetilde{\mathcal{A}}$ rather than the codebook $\mathcal{A}$ and train the protein decoder to reconstruct the embedding $\widetilde{\mathbf{e}}_{z_m}$ as $\widetilde{\mathbf{x}}_m$. Finally, the learning objective of Masked Codebook Modeling (MCM) is defined as

$$\mathcal{L}_{\text{MCM}} = \frac{1}{|\mathcal{V}_{[\mathcal{M}]}|} \sum_{v_m \in \mathcal{V}_{[\mathcal{M}]}} \left( 1 - \frac{\mathbf{x}_m^\top \widetilde{\mathbf{x}}_m}{\|\mathbf{x}_m\| \cdot \|\widetilde{\mathbf{x}}_m\|} \right)^\gamma, \text{where } \mathcal{V}_{[\mathcal{M}]} = \left\{ v_m \mid \widetilde{\mathbf{e}}_{z_m} \in \mathcal{M} \right\}, \tag{6}$$

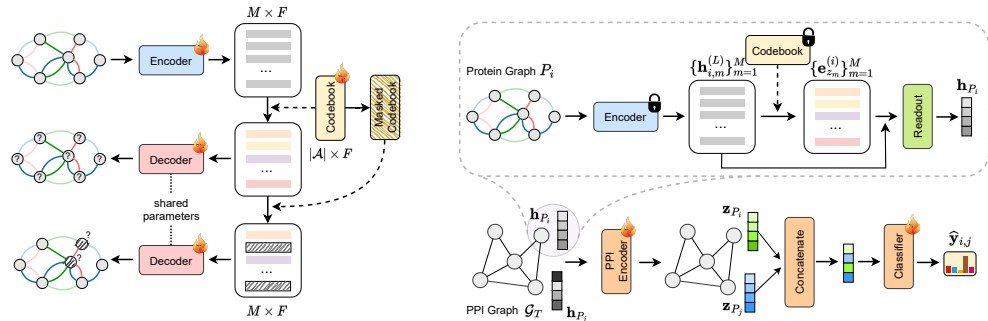

(a) Workflow for pre-training  (b) Microenvironment-aware protein embedding for PPI prediction

Figure 3: Illustration of pre-training the encoder and codebook for efficient PPI prediction, where the flame and lock icons indicate that the module is optimizable or parameter-frozen, respectively.

where the scaling factor $\gamma \geq 1$ is a hyper-parameter that adjusts the weight of each sample based on the reconstruction error (Hou et al., 2022). Finally, the total loss function can be defined as

$$\mathcal{L}_{\text{Pre}} = \mathcal{L}_{\text{VQ}} + \eta \mathcal{L}_{\text{MCM}}, \tag{7}$$

where $\eta$ is a hyper-parameter to balance the two losses.

### 4.4 EFFICIENT AND EFFECTIVE PPI PREDICTION

With the learned codebook $\mathcal{A}$, we can reuse it as an off-the-shelf tool to encode all proteins into microenvironment-aware protein embeddings. Specifically, we can directly map the microenvironment of the $m$-th residue of protein $P_i$ to the embedding $\mathbf{e}_{z_m}^{(i)}$, concate it with $\mathbf{h}_{i,m}^{(L)}$, and adopt a READOUT$(\cdot)$ operation to get the protein representation $\mathbf{h}_{P_i}$ of protein $P_i$ and freeze it, as follows

$$\mathbf{h}_{P_i} = \text{READOUT}\left(\left\{\left[\mathbf{e}_{z_m}^{(i)}\|\mathbf{h}_{i,m}^{(L)}\right] \mid v_m \in \mathcal{V}_P^{(i)}\right\}\right), \quad \text{where } 1 \leq i \leq N. \tag{8}$$

With the learned protein embeddings $\{\mathbf{h}_{P_i}\}_{i=1}^N$ as node features in the PPI graph $\mathcal{G}_T = (\mathcal{V}_T, \mathcal{E}_T)$, we adopt Graph Isomorphism Network (GIN) (Xu et al., 2018) as the PPI encoder, which update the node representations with a learnable parameter $\epsilon^{(l)}$ at $l$-th layer ($1 \leq l \leq L_s$), defined as:

$$\mathbf{z}_i^{(l)} = g^{(l)}\left(\left(1 + \epsilon^{(l)}\right) \cdot \mathbf{z}_i^{(l-1)} + \sum_{e_{i,j} \in \mathcal{E}_T} \mathbf{z}_j^{(l-1)}\right), \quad \text{where } 1 \leq i \leq N, \tag{9}$$

where $\mathbf{z}_i^{(0)} = \mathbf{h}_{P_i}$, and $g^{(l)}$ is a linear transformation of the $l$-th layer. Finally, we combine the updated protein embeddings $\mathbf{z}_{P_i} = \mathbf{z}_i^{(L_s)}$ and $\mathbf{z}_{P_j} = \mathbf{z}_i^{(L_s)}$ by dot product and use a fully connected layer (FC) as the PPI classifier to predict their interaction type $\widehat{\mathbf{y}}_{ij} = \text{FC}(\mathbf{z}_{P_i} \cdot \mathbf{z}_{P_i})$. Given a training set $\mathcal{D}_L = (\mathcal{X}_L, \mathcal{Y}_L)$, we train the PPI encoder and classifier via a binary cross-entropy, defined as

$$\mathcal{L}_{\text{PPI}} = \frac{1}{|\mathcal{X}_L|} \sum_{(P_i, P_j) \in \mathcal{X}_L} -\frac{1}{C} \sum_{c=1}^C \left(\mathbf{y}_{i,j}^c \log \sigma(\widehat{\mathbf{y}}_{ij}^c) + \left(1 - \mathbf{y}_{ij}^c\right) \log \left(1 - \sigma(\widehat{\mathbf{y}}_{ij}^c)\right)\right), \tag{10}$$

where $\sigma = \text{sigmoid}(\cdot)$ is the activation function, $\mathbf{y}_{i,j} \in [0,1]^C$ is the ground-truth interaction type between protein $P_i$ and protein $P_j$, and $C$ is the category number of PPI interaction types. Due to space limitations, the time complexity analysis and pseudo-code are placed in **Appendix E&F**.

## 5 EXPERIMENTS

**Datasets.** Extensive experiments are conducted on three datasets, namely STRING, SHS27k, and SHS148k. The STRING dataset contains 1,150,830 PPI entries of Homo sapiens from the STRING database (Szklarczyk et al., 2019), covering 14,952 proteins and 572,568 interactions. Furthermore, we follow Chen et al. (2019) to randomly select proteins with more than 50 amino acids and less than 40% sequence identity from the Homo sapiens subset of STRING to generate two more challenging datasets, SHS27k and SHS148k, which contain 16,912 and 99,782 entries of PPIs, respectively. Moreover, we apply Alphafold2 (Jumper et al., 2021) to predict the 3D structures of all protein sequence data. A detailed description of these three datasets is available in **Appendix A**. Besides, we adopt three partition algorithms proposed by Lv et al. (2021), including Random, Breath-First Search (BFS), and Depth-First Search (DFS) to split the training, validation, and test sets. To better evaluate generalizability, we further divide the test data into three subsets based on whether or not

Table 1: Performance of various methods (w/ and w/o additional pre-training data) over different datasets and partitions, where **bold** and underline denote the best and second metrics, respectively.

| Method | Pre-training Dataset (Size) | SHS27k | | | SHS148k | | | STRING | | | Ave. Rank |
|---|---|---|---|---|---|---|---|---|---|---|---|
| | | Random | DFS | BFS | Random | DFS | BFS | Random | DFS | BFS | |
| DPPI | - | 70.45 | 43.69 | 43.87 | 76.10 | 51.43 | 50.80 | 92.49 | 63.41 | 54.41 | 11.22 |
| DNN-PPI | - | 75.18 | 48.90 | 51.59 | 85.44 | 56.70 | 54.56 | 81.91 | 61.34 | 51.53 | 10.89 |
| PIPR | - | 79.59 | 52.19 | 47.13 | 88.81 | 61.38 | 58.57 | 93.68 | 64.97 | 53.80 | 9.89 |
| GNN-PPI | - | 83.65 | 66.52 | 63.08 | 90.87 | 75.34 | 69.53 | 94.53 | 84.28 | 75.69 | 8.11 |
| SemiGNN-PPI | - | 85.57 | 69.25 | 67.94 | 91.40 | 77.62 | 71.06 | 94.80 | 84.85 | 77.10 | 6.44 |
| HIGH-PPI | - | 86.23 | 70.24 | 68.40 | 91.26 | 78.18 | 72.87 | - | - | - | 5.83 |
| MAPE-PPI | - | **88.91** | **71.98** | **70.38** | **92.38** | **79.45** | **74.76** | **96.12** | **86.50** | **78.26** | 3.44 |
| ProBERT | BFD (2.1B) | 84.52 | 68.85 | 65.10 | 90.90 | 74.76 | 70.51 | 94.20 | 83.32 | 74.48 | 8.11 |
| ESM-1b | UniRef50 (24M) | 86.10 | 70.69 | 69.56 | 92.14 | 79.64 | 72.20 | 94.49 | 85.21 | 77.49 | 5.33 |
| KeAP | ProteinKG25 (5M) | 88.49 | 72.38 | 70.66 | 92.70 | 80.20 | 73.84 | 95.79 | 86.58 | **78.70** | 2.78 |
| GearNet-Edge | AlphaFoldDB (805K) | 89.36 | 72.06 | 71.42 | 92.88 | 79.84 | 74.43 | 96.33 | 85.96 | 78.24 | 2.78 |
| MAPE-PPI | CATH 4.2 (42K) | **90.42** | **73.15** | **72.47** | **93.39** | **81.12** | **75.62** | **96.90** | **87.08** | 78.59 | 1.11 |

two proteins have been seen in the training data, including (1) *BS*: both have been seen; (2) *ES*: either one proteins has been seen; and (3) *NS*: neither one has seen. In practice, the BFS and DFS partitions are more challenging, because they contain only ES and NS subsets in the test data.

**Evaluation Metrics and Hyperparameters.** Since the different PPI types are very imbalanced in the datasets used, micro-F1 may be preferred over macro-F1 as a metric to evaluate the performance of the multi-label PPI type prediction. Unlike previous studies (Lv et al., 2021; Zhao et al., 2023; Gao et al., 2023b) that selected 80% and 20% of the PPIs for training and testing and reported the best micro-F1 on the test set, we split the PPIs into the training (60%), validation (20%), and testing (20%) for all baselines. Then, we select the model that performs the best on the validation set to evaluate the micro-F1 scores of the test data. Beisdes, we run each set of experiments five times with different random seeds, and report the average performance. Due to space limitations, we defer the implementation details and the hyperparameters for each dataset and data partition to **Appendix B**.

**Baselines.** Most previous work on PPI type prediction has little to do with protein pre-training, i.e., they train only on downstream labeled data without leveraging unlabeled data. Following this line, we compare MAPE-PPI with DPPI (Hashemifar et al., 2018), DNN-PPI (Li et al., 2018), PIPR (Chen et al., 2019), GNN-PPI (Lv et al., 2021), SemiGNN-PPI (Zhao et al., 2023), and HIGH-PPI (Gao et al., 2023b), among which only HIGH-PPI is structure-based and the rest are sequence-based. Moreover, we also explore how protein pre-training on additional unlabeled data influences PPI prediction, by considering four pre-trained models, including ProBERT (Elnaggar et al., 2020), ESM-1b (Rives et al., 2021), GearNet-Edge (Zhang et al., 2022), and KeAP (Zhou et al., 2022).

## 5.1 PERFORMANCE COMPARISION

**Benchmark Comparison.** We compare the performance of MAPE-PPI with other baselines (w/ and w/o additional pre-training data) on three datasets under three data divisions in Tab. 1. For all pre-trained models, we freeze their output protein embeddings as the node features in the PPI graph and then make PPI predictions using the same PPI encoder and classifiers as in this paper for a fair comparison. Another way to utilize the pre-trained models is to fine-tune them, but we evaluate their computational efficiency in **Appendix C** and find it to be too computationally heavy. We can make three important observations from Tab. 1: (1) For the models trained only on downstream labeled data, both two structure-based models, HIGH-PPI and MAPE-PPI, outperform other sequence-based baselines, but HIGH-PPI still lags far behind MAPE-PPI by a large margin. (2) As for the two models pre-trained on protein sequences, ProBERT and ESM-1b cannot even match vanilla MAPE-PPI on a few datasets, which is trained without any additional unlabeled data. (3) Due to the large amount of structural and knowledge graph data used for pre-training, GearNet-Edge and KeAP outperform sequence-based ProBERT and ESM-1b, but still lag behind MAPE-PPI, which is pre-trained using only 42k unlabeled sequence-structure pairs from the CATH 4.2 dataset. More experimental results with other evaluation metrics, e.g., AUPR, are reported in **Appendix D**.

**Generalization Analysis.** To further evaluate the generalizability, we report the performance of MAPE-PPI and other baselines in Tab. 2 across the proportions of BS, ES, and BS subsets in the test set under three data divisions, Random, DFS, and BFS, on the SHS27k dataset. It can be seen that all methods perform significantly worse on the ES and NS subsets than on the BS subset, regardless of the data partition. Therefore, BFS and DFS partitions are more challenging than the Random partition because they contain only ES and NS subsets, whereas most of the samples in the Random partition are in the BS subset. More importantly, the performance gains of MAPE-PPI over other

Table 2: Proportions and performance comparison of BS, ES, and NS subsets on the SHS27k dataset.

| Method | Random Partition | | | | DFS Partition | | | BFS Partition | | |
|---|---|---|---|---|---|---|---|---|---|---|
| | BS (96.18%) | ES (3.77%) | NS (0.05%) | Ave. | ES (86.51%) | NS (13.49%) | Ave. | ES (73.42%) | NS (26.58%) | Ave. |
| PIPR | $89.59_{\pm0.59}$ | $72.83_{\pm0.74}$ | $54.54_{\pm1.24}$ | 88.94 | $62.83_{\pm1.23}$ | $54.57_{\pm1.47}$ | 61.72 | $61.72_{\pm1.51}$ | $51.27_{\pm1.31}$ | 58.94 |
| GNN-PPI | $91.34_{\pm0.41}$ | $74.58_{\pm0.85}$ | $55.54_{\pm0.74}$ | 90.69 | $76.71_{\pm1.07}$ | $67.10_{\pm1.51}$ | 75.41 | $72.64_{\pm1.34}$ | $60.53_{\pm1.45}$ | 69.41 |
| SemiGNN-PPI | $92.13_{\pm0.37}$ | $75.43_{\pm0.69}$ | $57.27_{\pm1.11}$ | 91.48 | $78.81_{\pm0.96}$ | $69.83_{\pm1.30}$ | 77.60 | $74.43_{\pm1.24}$ | $62.12_{\pm1.29}$ | 71.16 |
| HIGH-PPI | $91.72_{\pm0.28}$ | $75.11_{\pm0.53}$ | $58.18_{\pm0.85}$ | 91.10 | $79.22_{\pm1.14}$ | $70.49_{\pm1.44}$ | 78.04 | $75.93_{\pm1.35}$ | $63.84_{\pm1.20}$ | 72.72 |
| MAPE-PPI | $92.77_{\pm0.27}$ | $76.24_{\pm0.58}$ | $60.70_{\pm0.94}$ | 92.13 | $80.48_{\pm0.89}$ | $72.26_{\pm1.24}$ | 79.37 | $77.85_{\pm1.08}$ | $67.15_{\pm1.13}$ | 75.05 |
| $\Delta_{\text{SemiGNN-PPI}}$ | +0.69 % | +1.07 % | +5.99 % | +0.71 % | +2.12 % | +3.48 % | +2.28 % | +4.59 % | +8.10 % | +5.47 % |
| $\Delta_{\text{HIGH-PPI}}$ | +1.12 % | +1.50 % | +4.33 % | +1.13 % | +1.59 % | +2.51 % | +1.70 % | +2.53 % | +5.18 % | +3.20 % |

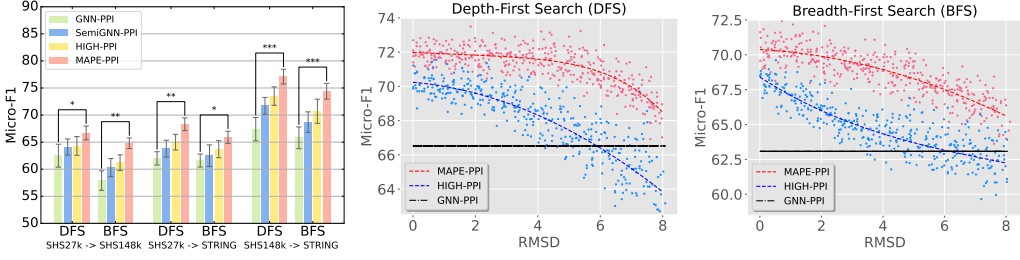

(a) Domain Generalization     (b) Robustness Evaluation in DFS     (c) Robustness Evaluation in BFS

Figure 4: (a) Generalization performance comparison of testing on unseen trainset-heterogenous test data under different data partitions. (bc) Robustness evaluation on protein 3D structures with different accuracy, measured by Root Mean Square Deviation (RMSD), on the SHS27k dataset.

baselines mainly come from the ES and NS subsets. For example, MAPE-PPI improves HIGH-PPI over 4.33% in the NS subset, but only 1.12% in the BS subset under the random partition.

## 5.2 EVALUATION OF DOMAIN GENERALIZATION AND ROBUSTNESS

**Domain Shifting.** To explore how domain shifting affects generalization, we transfer a model trained on a smaller dataset to a larger dataset. For example, a model trained on the SHS27k dataset will encounter proteins with fewer amino acids on the STRING dataset. More importantly, unlike the BFS and DFS divisions on the SHS27k dataset, the vast majority (more than 80%) of PPIs on test data are located in the NS subset when migrating to STRING. We report the performance of four methods in three domain shift settings in Fig. 4(a), from which we can see that MAPE-PPI outperforms other baselines across all settings, especially when migrating from SHS148k to STRING.

**Robustness.** To evaluate the robustness of MAPE-PPI to the accuracy of protein 3D structures, we randomly perturb the 3D coordinates of protein structures predicted by AlphaFold2 (Jumper et al., 2021) (termed AF2 structures) to obtain perturbed 3D structures with different RMSDs to the AF2 structures. We report the performance of HIGH-PPI and MAPE-PPI when testing on protein structures of different RMSDs under two partitions in Fig. 4(b) and Fig. 4(c), as well as the performance of a sequence-based method, GNN-PPI. We observe that MAPE-PPI outperforms HIGH-PPI across all structural perturbations. Moreover, HIGH-PPI is significantly more affected by structural perturbations than MAPE-PPI, and even inferior to GNN-PPI, when RMSD is overly large.

## 5.3 EVALUATION ON LEARNED MICROENVIRONMENT CODEBOOK

**Embedding Visualization.** To provide an intuitive understanding, we visualize in Fig. 5(a) the embeddings of four microenvironment codes $\{\mathbf{e}_{113}, \mathbf{e}_{193}, \mathbf{e}_{348}, \mathbf{e}_{509}\} \in \mathcal{A}$, as well as residue embeddings $\{\mathbf{h}_m^{(L)}\}_{z_m=n}$ that are mapped to each microenvironment code $\mathbf{e}_n$. According to the figure, the residue embeddings are tightly centered around the corresponding microenvironment codes, respectively, and there are clear boundaries between different microenvironment codes and their residue embeddings. This suggests that codebook updating is essentially equivalent to performing clustering on the residual embeddings, and microenvironment codes can be regarded as the clustering centers.

**Distribution of Amino Acids (AA).** We explore the distribution of amino acids in depth from two different aspects. First, we present the distribution of amino acid types corresponding to each microenvironment code in Fig. 5(b). We find that different microenvironment codes tend to encode the local environments of different amino acids. For example, both microenvironment codes 113 and 193 primarily encode the local environment around amino acid $L$, but they differ in specific AA distribution. Furthermore, we also count all the AA that are primarily encoded by each microenvironment code, calculate the distribution of AA in the real-world SHS27k dataset, and plot them in a

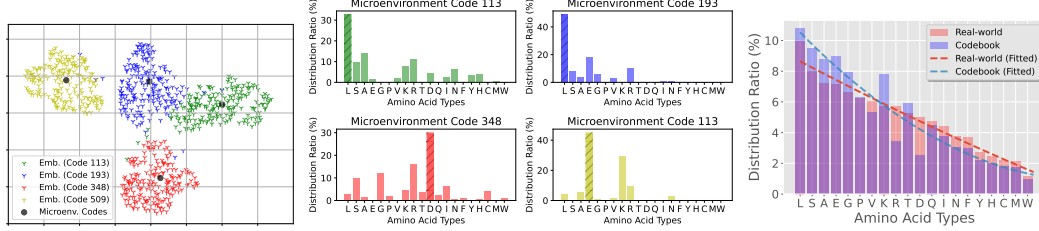

(a) Visualization by UMAP    (b) Microenvironment-level AA Distribution    (c) Code-level AA Distribution

Figure 5: (a) Visualization (by UMAP) of the embeddings of four microenvironment codes and corresponding residues on the SHS27k dataset. (b) Distribution of amino acids within each microenvironment code on the SHS27k. (c) Distribution of amino acids primarily encoded by each microenvironmental code, as well as the distribution of amino acids in the real-world SHS27k dataset.

histogram in Fig. 5(c). Based on the figure, we conclude that the distribution of amino acid types in our codebook is similar to that of the real world, i.e., *the more of a specific amino acid that exists in nature, the more codes in the codebook will be allocated to encode its local environment.*

### 5.4 ABLATION STUDY AND HYPERPARAMETRIC SENSITIVITY

**Ablation Study.** To further explore how masked modeling, vector quantization, and fine-tuning influence performance, we compare vanilla MAPE-PPI with five other schemes: *(A)* No Masking: the model is trained without masked modeling; *(B)* w/ MLM: the model is trained with masked input features $\{\mathbf{x}_m\}_{m=1}^M$ instead of MCM; *(C)* w/ MHM: train with masked hidden codes $\{z_m\}_{m=1}^M$ instead of MCM; *(D)* w/o VQ: the model is trained without vector quantization by directly using residue embeddings $\{\mathbf{h}_m^{(L)}\}_{m=1}^M$; and *(E)* fine-tune (rather than freeze) the protein encoder by downstream supervision. We can observe from the results in Tab. 3 that (1) Masked modeling plays an important role, but the performance gains of MLM and MHM are much less than our proposed MCM. (2) The removal of vector quantization leads to significant performance drops, which illustrates the effectiveness of the codebook and vector quantization. (3) Fine-tuning the protein encoder helps improve performance, but at the cost of a huge computational burden, as shown in **Appendix C**.

Table 3: Ablation study on various modules on three datasets.

| Method | SHS27k | | SHS148k | | STRING | | Average |
|---|---|---|---|---|---|---|---|
| | DFS | BFS | DFS | BFS | DFS | BFS | |
| MAPE-PPI (w/ MCM) | 71.98 | 70.38 | 79.45 | 74.76 | 86.50 | 78.26 | 76.89 |
| No Masking | 70.11 | 68.60 | 78.02 | 72.93 | 84.90 | 76.99 | 75.26 (-2.12 %) |
| w/ MIM | 70.85 | 69.25 | 78.39 | 73.89 | 85.46 | 77.20 | 75.84 (-1.37 %) |
| w/ MHM | 70.44 | 68.28 | 77.70 | 73.06 | 84.02 | 76.39 | 74.98 (-2.48 %) |
| w/o VQ | 69.52 | 68.43 | 78.25 | 72.69 | 85.10 | 76.48 | 75.08 (-2.35 %) |
| Fine-tuned Encoder | 72.56 | 70.81 | 79.94 | 74.60 | 87.12 | 78.09 | 77.19 (+0.39 %) |

**Hyperparametric Sensitivity.** We investigate the sensitivity of the codebook size $|\mathcal{A}|$ and mask ratio $|\mathcal{M}|/|\mathcal{A}|$ on the SHS27k dataset. As shown in Tab. 4, codebook size does influence model performance; a codebook size that is set too small, e.g., 64 or 128, cannot cover the diversity of microenvironments, resulting in poor performance. Although setting $|\mathcal{A}|$ to 2048 can sometimes outperform 512 or 1024, the superiority is not significant. Therefore, we set the codebook size as 512 or 1024 in this paper for a smaller computational burden. Besides, we find that a mask ratio of 0.15 usually yields good performance, since a small mask ratio, e.g. 0.05, weakens the contribution of masked codebook modeling, while a large mask ratio, e.g. 0.35, hinders the codebook learning.

Table 4: Performance with various coodebook sizes $|\mathcal{A}|$ and mask ratios $\frac{|\mathcal{M}|}{|\mathcal{A}|}$ on the SHS27k dataset.

| $|\mathcal{A}|$ | 64 | 128 | 256 | 512 | 1,024 | 2,048 | $|\mathcal{M}|/|\mathcal{A}|$ | 0.05 | 0.10 | 0.15 | 0.20 | 0.25 | 0.35 |
|---|---|---|---|---|---|---|---|---|---|---|---|---|---|
| DFS | 66.57 | 68.46 | 70.35 | **71.98** | 71.53 | 71.81 | DFS | 70.67 | 71.98 | **72.34** | 71.10 | 69.43 | 67.71 |
| BFS | 65.32 | 67.24 | 69.14 | 70.13 | **70.38** | 69.95 | BFS | 69.86 | 69.59 | **70.38** | 70.24 | 68.30 | 66.25 |

## 6 CONCLUSION

In this paper, we unify microenvironment discovery and microenvironment-aware protein embedding in an end-to-end framework. Besides, we propose Masked Codebook Modeling that directly masks the codebook instead of input features to capture the dependencies between microenvironments. Extensive experiments have shown that MAPE-PPI can scale to million-level PPI prediction with a superiority in both effectiveness and computational efficiency compared to state-of-the-art baselines. Limitations still exist, for example, how to extend MAPE-PPI to predict the interaction interfaces or conformations among multiple proteins is still a promising direction for future work.

# 7 ACKNOWLEDGMENTS

This work was supported by National Key R&D Program of China (No. 2022ZD0115100), National Natural Science Foundation of China Project (No. U21A20427), and Project (No. WU2022A009) from the Center of Synthetic Biology and Integrated Bioengineering of Westlake University.

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

# APPENDIX

## A. DATASETS

We conduct extensive experiments on three public PPI datasets, namely STRING, SHS27k, and SHS148k. The STRING dataset contains 1,150,830 entries of PPIs for Homo sapiens from the STRING database (Szklarczyk et al., 2019), covering 14,952 proteins and 572,568 interactions. Each protein-protein interaction is annotated with at least one of **seven types**, i.e., Activation, Binding, Catalysis, Expression, Inhibition, Post-translational modification (Ptmod), and Reaction. In this paper, *"one interaction + one annotation" is referred to as one entry of PPI data*. Moreover, SHS27k and SHS148k are two subsets of STRING, where proteins with more than 50 amino acids and less than 40% sequence identity are included. The SHS27k dataset contains 16,912 entries of PPIs, covering 1,663 proteins and 7,401 interactions. The SHS148k dataset contains 99,782 entries of PPIs, covering 5,082 proteins and 43,397 interactions. Furthermore, we adopt three partition algorithms, including Random, Breath-First Search (BFS), and Depth-First Search (DFS) (Lv et al., 2021) to split each dataset into the training, validation, and test sets with a split ratio of 6: 2: 2.

## B. IMPLEMENTATION DETAILS AND HYPERPARAMETERS

The following hyperparameters are set the same for all datasets and partitions: PPI encoder (GIN) with layer number $L_s = 2$ and hidden dimension 1024, learning rate $lr = 0.001$, weight decay $decay = 1e - 4$, loss weight $\beta = 0.25$, pre-training epoch $E_{pre} = 50$, PPI training epoch $E = 500$, thresholds $d_s = 2$, $d_r = 10\mathring{A}$, and neighbor number $K = 5$. The other dataset-specific hyperparameters are determined by an AutoML toolkit NNI with the hyperparameter search spaces as: protein encoder with layer number $L = \{4, 5\}$ and hidden dimension $F = \{128, 256\}$, codebook size $|\mathcal{A}| = \{256, 512, 1024\}$, mask ratio $\frac{|\mathcal{M}|}{|\mathcal{A}|} = \{0.1, 0.15, 0.2\}$, scaling factor $\gamma = \{1, 1.5, 2.0\}$, and loss weight $\eta = \{0.5, 1.0\}$. For a fairer comparison, the model with the highest validation accuracy is selected for testing. The best hyperparameter choices of each dataset and data partition are available at https://github.com/LirongWu/MAPE-PPI. Moreover, the experiments on both baselines and our approach are implemented based on the standard implementation using the PyTorch 1.6.0 with Intel(R) Xeon(R) Gold 6240R @ 2.40GHz CPU and 8 NVIDIA A100 GPUs.

## C. EFFICIENCY VS. EFFECTIVENESS

For a fair comparison, we take the protein embeddings output by the pre-trained protein encoder as node features in the PPI graph and then use the PPI encoder and classifier proposed in this paper to make PPI predictions. We consider the following two paradigms for utilizing pre-trained models:

- *FT*: fine-tuning the pre-trained protein encoder and training the PPI encoder and classifier;
- *FREEZE*: freezing the pre-trained encoder and training the PPI encoder and classifier.

We report the micro-F1 scores and training time for both two training paradigms as well as those models trained without additional pre-training data on the SHS27k dataset (Random partition) in Fig. A1. Since the pre-training time for the pre-trained models differs significantly and is much higher than the time required to train the PPI model, we omit the pre-training time in Fig. A1 and only report the time taken to fine-tune or freeze the encoder and train the PPI model. Several important observations can be derived from Fig. A1: (1) For those models trained only on downstream labeled data, although HIGH-PPI performs better than the other sequence-based baselines, approaching our MPAE-PPI, it requires nearly $50\times$ more training time than MPAE-PPI. (2) For models that are pre-trained with additional unlabeled data, while they usually perform better in the *FT* setting than in the *FREEZE* setting, this comes at the cost of an extremely high computational burden. For example, even on the SHS27k dataset containing only 16,912 PPI data, ESM-1b takes more than 10 hours to fine-tune (*vs.* only 300s for the *FREEZE* setting), which makes it hard to scale to prediction of millions of PPIs. *Therefore, considering the unaffordable computational overhead of fine-tuning pre-trained models on large-scale PPI datasets, we only consider the FREEZE setting in the main paper.* (3) Considering both efficiency and effectiveness, MAPE-PPI makes the best trade-off, regardless of w/ or w/o additional pre-training data, and in the *FT* or *FREEZE* setting.

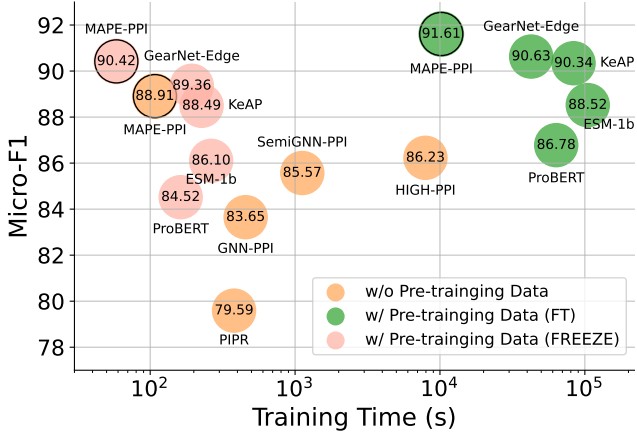

Figure A1: Effectiveness (micro-F1 scores) and efficiency (training time) for models (two training paradigms) w/ and w/o additional unlabeled data on the SHS27k dataset in the Random setting.

## D. PERFORMANCE COMPARISON WITH OTHER EVALUATION METRICS

We compare MAPE-PPI to GNN-PPI, SemiGNN-PPI, and HIGH-PPI in Tab. A1, using *Area Under the Precision-Recall curve* (AUPR) scores rather than micro-F1 scores as the metrics. It can be seen that MAPE-PPI achieves the best performance under all three data divisions on three datasets.

Table A1: Performance (measured by AUPR, %) of MAPE-PPI and other baselines over different datasets and partitions, where **bold** and underline denote the best and second metrics, respectively.

| Method | SHS27k | | | SHS148k | | | STRING | | | Average |
|---|---|---|---|---|---|---|---|---|---|---|
| | Random | DFS | BFS | Random | DFS | BFS | Random | DFS | BFS | |
| GNN-PPI | 86.44 | 62.72 | 55.19 | 91.57 | 74.58 | 70.91 | 89.83 | 79.52 | 66.06 | 75.20 |
| SemiGNN-PPI | 88.70 | 69.29 | 67.15 | 92.45 | 77.82 | 72.84 | 90.45 | 80.10 | 68.27 | 78.56 |
| HIGH-PPI | 89.20 | 70.98 | 68.27 | 92.21 | 79.10 | 74.22 | - | - | - | - |
| MAPE-PPI (ours) | **90.18** | **72.79** | **70.90** | **93.09** | **80.71** | **76.13** | **91.51** | **81.33** | **70.64** | 80.81 |
| $\Delta_{\text{SemiGNN-PPI}}$ | +1.67 % | +5.05 % | +5.58 % | +0.69 % | +3.71 % | +4.52 % | +1.17 % | +1.54 % | +3.47 % | +2.86 % |
| $\Delta_{\text{HIGH-PPI}}$ | +1.10 % | +2.55 % | +3.85 % | +0.95 % | +2.04 % | +2.57 % | - | - | - | - |

## E. TRAINING TIME COMPLEXITY ANALYSIS

The training time complexity of MAPE-PPI comes from four parts: (1) protein encoding $\mathcal{O}(N|\mathcal{V}_P| + N|\mathcal{E}_P|)$; (2) vector quantization $\mathcal{O}(N|\mathcal{V}_P| \cdot |\mathcal{A}|)$; (3) protein decoding (reconstruction) $\mathcal{O}(N|\mathcal{V}_P| + N|\mathcal{E}_P|)$; and (4) PPI prediction $\mathcal{O}(|\mathcal{E}_T|)$, where $N$ is the number of proteins, $|\mathcal{E}_T|$ is the number of edges in the PPI graph $\mathcal{G}_T = (\mathcal{V}_T, \mathcal{E}_T)$, $|\mathcal{V}_P| = \frac{1}{N} \sum_{i=1}^{N} |\mathcal{V}_P^{(i)}|$ and $|\mathcal{E}_P| = \frac{1}{N} \sum_{i=1}^{N} |\mathcal{E}_P^{(i)}|$, are the average number of nodes and edges in protein graphs $\{\mathcal{G}_P^{(i)} = (\mathcal{V}_P^{(i)}, \mathcal{E}_P^{(i)})\}_{i=1}^{N}$. The total training time complexity of MAPE-PPI is $\mathcal{O}(N(|\mathcal{V}_P| \cdot |\mathcal{A}| + |\mathcal{E}_P|) + |\mathcal{E}_T|)$. Besides, the time complexity of HIGH-PPI for protein encoding and PPI prediction is $\mathcal{O}(N(|\mathcal{V}_P| + |\mathcal{E}_P|))$ (without vector quantization and protein decoding) and $\mathcal{O}(|\mathcal{E}_T|)$, respectively. However, since HIGH-PPI jointly performs protein encoding and PPI prediction in an end-to-end manner (while MAPE-PPI decouples the two), its total training time complexity is the product of the two, that is $\mathcal{O}(N(|\mathcal{V}_P| + |\mathcal{E}_P|) \cdot |\mathcal{E}_T|)$, which is much higher than that of MAPE-PPI, since we have $|\mathcal{V}_P| < |\mathcal{A}| \ll |\mathcal{E}_P| < |\mathcal{E}_T|$ in practice.

## F. PSEUDO CODE

The pseudo-code of the proposed MAPE-PPI framework is summarized in Algorithm 1.

---

**Algorithm 1** Microenvironment-aware Protein Embedding for PPI Prediction (MAPE-PPI)

---

**Input:** Proteins $\mathcal{P} = \{P_i\}_{i=1}^N$, PPIs set $\mathcal{X}$, pre-training epoch $E_{pre}$, and training epoch $E$.

1: Randomly initializing the parameters of protein encoder $f_e$, decoder $f_d$, and codebook $\mathcal{A}$.
2: # *Pre-training*
3: **for** epoch $\in \{1, 2, \cdots, E_{pre}\}$ **do**
4:     Encoding residue embeddings $\{\mathbf{h}_m^{(L)}\}_{m=1}^M$ for each protein by the encoder $f_e$ in Eq. (2);
5:     Performing vector quantization by the codebook $\mathcal{A}$ in Eq. (3);
6:     Reconstructing the input from discrete codes $\{z_m\}_{m=1}^M$ by the decoder $f_d$;
7:     Masking the codebook by Eq. (5) and reconstructing the input from the masked $\{\widetilde{\mathbf{e}}_{z_m}\}_{m=1}^M$;
8:     Optimizing encoder $f_e$, decoder $f_d$, and codebook $\mathcal{A}$ by minimizing the loss in Eq. (7).
9: **end for**
10: # *PPI Prediction*
11: Freezing the parameters of encoder $f_e$ and codebook $\mathcal{A}$.
12: Randomly initializing the parameters of PPI encoder $g_{\text{enc}}$ and PPI classifier $g_{\text{cla}}$.
13: **for** epoch $\in \{1, 2, \cdots, E\}$ **do**
14:     Encoding all proteins into microenvironment-aware protein embeddings by Eq. (8);
15:     Updating node (protein) embeddings on the PPI graph by the PPI encoder $g_{\text{enc}}$ in Eq. (9);
16:     Predicting unknown PPI types by the PPI classifier $g_{\text{cla}}$.
17:     Optimizing PPI encoder $g_{\text{enc}}$ and PPI classifier $g_{\text{cla}}$ by minimizing the objectivein in Eq. (10).
18: **end for**
19: **return** Predicted PPI types $\mathcal{Y}_U$, protein encoder $f_e$, and microenvironment codebook $\mathcal{A}$.

---

