# OpenReview forum: "MAPE-PPI: Towards Effective and Efficient Protein-Protein Interaction Prediction via Microenvironment-Aware Protein Embedding"
_ICLR.cc/2024/Conference — ICLR 2024 spotlight_

### Official Review · Reviewer_XV9S · 2023-10-27

**Soundness:** 3 good
**Presentation:** 3 good
**Contribution:** 4 excellent
**Rating:** 8
**Confidence:** 5

**Summary:**

This paper starts with the biological concept of microenvironment and provides a new definition of it from a deep learning perspective. The authors adopt a variant of VQ-VAE to well formulate microenvironment discovery as a codebook learning problem, and propose a novel pre-training strategy specialized for microenvironments, i.e., capturing the dependencies between different microenvironments by randomly masking the codebooks and reconstructing the inputs. Extensive experiments on various datasets, metrics, and data partitions have adequately demonstrated the advantages of the proposed approach in terms of efficiency, effectiveness, generalization, and robustness. Overall, this paper has solid contributions and is expected to have a great impact on protein engineering, especially protein representation learning and PPI prediction.

**Strengths:**

(1) Starting from the perspective of microenvironments is a bright spot, compared to previous works on residue-level and protein-level modeling. More importantly, the definition of the microenvironment takes into account both sequence and structural contextual information, constructing it as a heterogeneous subgraph of the protein graph, which is well suited to existing deep learning models.

(2) One of the main contributions of this paper is the formulation of microenvironment discovery as a codebook learning problem that greatly extends the limited vocabulary of (20 types) amino acids while taking into account the structural context of proteins. To the best of my knowledge, this is the first work to implement both microenvironment discovery and embedding in a computational way.

(3) A novel pre-training approach for codebook masking modeling has been specifically proposed to capture dependencies between different microenvironments in the learned codebook.

(4) MAPE-PPI inherits the high efficiency of sequence-based methods, while enjoying the structure awareness of structure-based methods, showing a good trade-off between efficiency and effectiveness.

(5) Adequate experiments, including a variety of datasets, metrics, and data partitions, covering aspects of effectiveness, efficiency, generalizability, robustness, ablation study, and visualization.

**Weaknesses:**

(1) This paper proposes masking codebook modeling as a pre-training task. Has this approach been used before in other domains (e.g., CV and NLP) or is it the originality of this paper? The authors are encouraged to explain more about the motivation for doing so and provide more experiments to demonstrate its advantages over previous existing masked modeling approaches.

(2) Why does MAPE-PPI freeze the protein encoder instead of fine-tuning it?

(3) This paper introduces validation sets for hyperparameter tuning and model selection in the data partition, which makes more sense than the previous data partition. The authors have also reproduced the results of HIGH-PPI using the same data partitioning as MAPE-PPI in Tables 1 and 2, right?

(4) Some typos need to be corrected.

**Questions:**

It would be better for the authors to explain more about the issues in the weaknesses part.

---

> ### Author Response · Authors · 2023-11-12
> **Response to Reviewer XV9S**
>
> Thanks for your insightful comments and constructive suggestions! We mark the contents that already existed (but were missed by reviewers) in red, and those revised or newly added contents in blue in the revised manuscript. Next, we will address the concerns you raised, one by one, as described below:
>
> ---
>
> **Q1**: This paper proposes masking codebook modeling as a pre-training task. Has this approach been used before in other domains (e.g., CV and NLP) or is it the originality of this paper? The authors are encouraged to explain more about the motivation and provide more experiments to demonstrate its advantages over existing masked modeling methods.
>
> **A1**: We'd like to highlight that Masked Codebook Modeling (MCM) is originally proposed in this paper and is one of our core contributions. We answer your question in three aspects:
>
> - **Novelty.** We appreciate a large amount of works on VAE and masked modeling for CV and NLP. However, MCM is not a simple extension of previous masked modeling methods to VAE-based codebook updating. One of the biggest differences between MCM and related work (including [1] as pointed out by Reviewer wZyc) has been explained in **Lines 139-141**, that
>
>     > "Following this line in this paper, but unlike previous works that mask input features or hidden codes [1] for *training a better image encoder or decoder*, we randomly mask the microenvironment codes in the codebook to capture their dependencies for *learning a better codebook*."
>     >
>
> - **Motivation.** The motivation behind MCM is that there are generally overlaps (dependencies) in the microenvironments of two residues that are adjacent in sequence or structure. To capture such microenvironmental dependencies in the codebook, we propose Masked Codebook Modeling (MCM) to mask the codebook $\mathcal{A}$ and then reconstruct the inputs based on the unmasked codes, aiming to learn a more expressive codebook $\mathcal{A}$ rather than a more powerful protein encoder. The above explanations have been described in **Lines 218-222**.
>
> - **Comparison.** We provide a comparison of MCM with two schemes, (1) Masked Input Modeling(MIM) that masks input features {$\mathbf{x}_m$}$\_{m=1}^M$ and (2) Masked Hidden Modeling (MHM) that masks hidden codes {$z_m$}$\_{m=1}^M$, in **Lines 345-351** and **Table 3** of the original manuscript. The reported results have demonstrated the superiority of MCM over MIM and MHM.
>
> [1] Peng Z, Dong L, Bao H, et al. Beit v2: Masked image modeling with vector-quantized visual tokenizers[J]. arXiv preprint arXiv:2208.06366, 2022.
>
> ---
>
> **Q2**: Why does MAPE-PPI freeze the protein encoder instead of fine-tuning it?
>
> **A2**: We have explained why MAPE-PPI freeze rather than fine-tune the protein encoder in **Lines 283-284** of the main paper and **Lines 562-569** of **Appendix C**. One main concern is the unaffordable computational overhead of fine-tuning the pre-trained encoder on large-scale PPI datasets. For example, even on the SHS27k dataset containing only 16,912 PPI data, ESM-1b takes more than 10 hours to fine-tune (vs. only ~300s for the FREEZE setting).
>
> ---
>
> **A3**: This paper introduces validation sets for hyperparameter tuning and model selection in the data partition, which makes more sense than the previous data partition. The authors have also reproduced the results of HIGH-PPI using the same data partitioning as MAPE-PPI in Tables 1 and 2, right?
>
> **Q3**: You are right. We split the PPIs into the training (60%), validation (20%), and testing (20%) for all baselines (including both MAPE-PPI and HIGH-PPI), as we explained in **Lines 263-266**.
>
> ---
>
> **A4**: Some typos need to be corrected.
>
> **Q4**: All typos have been corrected in the revised manuscript.
>
> ---
>
> In light of these responses, we hope we have addressed your concerns. If we have left any notable points of concern unaddressed, please do share and we will attend to these points. We sincerely hope that you can appreciate our efforts on responses and revisions and thus raise your score.

---

> > ### Comment · Reviewer_XV9S · 2023-11-23
> >
> > Thank you for the response. I decided to keep my positive score but raise confidence to 5.

---

### Official Review · Reviewer_4o6m · 2023-10-31

**Soundness:** 3 good
**Presentation:** 3 good
**Contribution:** 3 good
**Rating:** 6
**Confidence:** 4

**Summary:**

In this paper, the authors proposed a new method, named MPAE-PPI, for protein-protein interaction prediction. Based on the experimental results, their method showed superiority, compared with other existing methods on several real data sets.

**Strengths:**

The author describes the fundamental algorithm well; and they seem to give all relevant information to understand and reproduce their algorithm.

The overall writing is satisfactory. The writing is fluent and clear and the ideas are easy to follow.

The proposed method is relative better than previous methods, which is not lack of significance.

**Weaknesses:**

(1) To make their results more convincing, they should compare their method with more latest structure-based methods.
(2) They just perform parameter sensitivity analysis on one dataset. It remains unclear whether the proposed method is sensitive to the hyper-parameters and how to setup the values in general cases.
(3) Lack of description of the details of the datasets.
(4) Lack of real world examples to demonstrate the effectiveness of their model.

**Questions:**

(1) To make their results more convincing, they should compare their method with more latest structure-based methods.
(2) They just perform parameter sensitivity analysis on one dataset. It remains unclear whether the proposed method is sensitive to the hyper-parameters and how to setup the values in general cases.
(3) Lack of description of the details of the datasets.
(4) Lack of real world examples to demonstrate the effectiveness of their model.

---

> ### Author Response · Authors · 2023-11-12
> **Response to Reviewer 4o6m**
>
> Thanks for your insightful comments and constructive suggestions! We mark the contents that already existed (but were missed by reviewers) in red, and those revised or newly added contents in blue in the revised manuscript. Next, we will address the concerns you raised, one by one, as described below:
>
> ---
>
> **Q1**: To make their results more convincing, they should compare their method with more latest structure-based methods.
>
> **A1**: HIGH-PPI is a method published in February 2023 in Nature Communications. To the best of our knowledge, it is the first and also the only structure-based method for protein-protein interaction type prediction. Unlike protein-protein docking and interaction site prediction, for which most works are structure-based, PPI type prediction is currently undergoing a paradigm shift from sequence-based to structure-based, and thus there is indeed not much work available for comparison. One of the values of our paper is that it promises to fill a gap in this area.
>
> ---
>
> **Q2**: They just perform parameter sensitivity analysis on one dataset. It remains unclear whether it is sensitive to the hyper-parameters and how to setup the values in general cases.
>
> **A2**: We have further evaluated the sensitivity of the codebook size $|\mathcal{A}|$ and mask ratio $\frac{|\mathcal{M}|}{|\mathcal{A}|}$ on two additional datasets, SHS148k and STRING, respectively. The detailed experimental results and analysis can be found in **Appendix G**, **Lines 589-593**. Besides, we have described in detail how we determined the hyperparameters in **Lines 266-267** of the main paper and **Lines 539-543** of **Appendix B**. Specifically, the hyperparameters are determined by an AutoML toolkit NNI with pre-defined hyperparameter search spaces. For a fair comparison, the model with the highest validation accuracy will be selected for testing.
>
> ---
>
> **Q3**: Lack of description of the details of the datasets.
>
> **A3**: Due to space limitations, we place a detailed description of the datasets in **Lines 523-534** of **Appendix A**. For more details, please refer directly to the STRING database [1].
>
> [1] Szklarczyk D, Gable A L, Lyon D, et al. STRING v11: protein–protein association networks with increased coverage, supporting functional discovery in genome-wide experimental datasets[J]. Nucleic acids research, 2019, 47(D1): D607-D613.
>
> ---
>
> **Q4**: Lack of real world examples to demonstrate the effectiveness of their model.
>
> **A4**: We do not get precisely what is meant by "real-world examples". If it refers to experimental designs, we have tried to make them consistent with real-world scenarios, including real-world Homo sapiens datasets and various data splits to evaluate real-world generalizations and robustness. If it refers to real-world experimental assays in a wet lab, we regret not including them in this paper, as they are quite expensive and time-consuming. As a conference on "representation learning", we are more concerned with the design of computational methods.
>
> If "real-world examples" are very important and our understanding is not right, please do not hesitate to contact us.
>
> ---
>
> In light of these responses, we hope we have addressed your concerns. If we have left any notable points of concern unaddressed, please do share and we will attend to these points. We sincerely hope that you can appreciate our efforts on responses and revisions and thus raise your score.

---

### Official Review · Reviewer_wZyc · 2023-11-04

**Soundness:** 3 good
**Presentation:** 3 good
**Contribution:** 2 fair
**Rating:** 3
**Confidence:** 4

**Summary:**

Summary.

This paper is dedicated to developing protein-protein interaction models. The authors point out the efficiency bottleneck of protein structure modeling. They introduce the microenvironments of an amino acid residue. Based on this, they propose microenvironment-aware protein embeddings for PPI prediction, which encode microenvironment into chemically meaningful discrete codes. The authors leverage this design to enable masked codebook modeling training. Experiments are conducted to scale PPI prediction with millions of PPIs.

**Strengths:**

Pros.

1. The figures are nice and informative.
2. Various PPI baselines are considered, including both sequence- and structure-based approaches.

**Weaknesses:**

Cons.

1. The authors mention that "the complexity of protein structure modeling hinders its application to large-scale PPI prediction", and emphasize it as an efficiency limitation. Why? It is straightforward to see that complexity modeling hinders accurate modeling. However, it is not clear how it hinders the scalability of modeling and why it is inefficient.
2. Important references are missing. (i) https://arxiv.org/pdf/2208.06366.pdf, which does almost the same codebook learning and masked codebook modeling in vision; (ii) https://www.nature.com/articles/s41586-022-04599-z, which define microenvironments of an amino acid residue and do masked "token" prediction. Based on this omitted reference, the innovation of this work is limited. Also, it is inappropriate to claim "propose... MCM", since both microenvironments and MCM are pre-defined in the literature.
3. What is the investigation of vocabulary redundancy? The authors create a sufficiently large vocabulary. Are they redundant? Or what is the relationship between vocabulary size or redundancy and archivable performance?
4. The efficiency claim is very confusing. "Extensive experiments show that MAPE-PPI can scale to PPI prediction with millions of PPIs" seems to suggest an inference efficiency. "predict unknown PPIs more efficiently and effectively" also seems to suggest an inference efficiency. However, in the introduction, the authors claim the training efficiency. Also, why the proposed design can enable efficiency? What are the computational and memory overheads of the sufficiently large vocabulary and two-stage training? More detailed time complexity analyses are needed.
5. Only a single level of sequence similarity is considered. More like 30%, 50%, 90% are encouraged. Meanwhile, as the main results in Table 1, multiple runs of experiments are encouraged to show the prediction stability.

**Questions:**

Refer to the weakness section.

---

> ### Author Response · Authors · 2023-11-12
> **Response to Reviewer wZyc - Part (1/2)**
>
> Thanks for your insightful comments! We mark the contents that already existed (but were missed by reviewers) in red, and those revised or newly added contents in blue in the revised manuscript. Next, we will address the concerns you raised, one by one, as described below:
>
> ---
>
> **Q1**: The authors mention that "the complexity of protein structure modeling hinders its application to large-scale PPI prediction", and emphasize it as an efficiency limitation. Why? It is not clear how it hinders the scalability of modeling and why it is inefficient.
>
> **A1**: The ambiguity of the term "complexity" may have led to some misunderstandings. The term "complexity" refers not to the complexity of the protein structure itself in this paper, but to the complexity of the algorithms that model (encode) the protein structure. The complexity of the protein structure does affect the modeling accuracy, but the algorithmic complexity may have a significant impact on the training efficiency. To clarify this point, we have made the following corrections and additions:
> - We have replaced “the complexity of protein structure modeling hinders …” with a more concise expression “the training inefficiency of existing structure-based methods hinders …”.
> - We provide a time-complexity comparison of the structure-based method (HIGH-PPI) with MAPE-PPI in **Appendix E**, **Lines 576-586**, which explains why HIGH-PPI is inefficient.
>
> ---
>
> **Q2**: Important references [1,2] are missing. Based on this omitted reference, the innovation of this work is limited.
>
> **A2**: The two papers [1,2] are relevant to our work, but not as important as you said. We sincerely hope that you do not conclude that our work is "limited innovation" just because these two papers share some similar concepts and terminology with us. We hope that the following comparisons will help you better appreciate our contributions and how we differ from [1,2]:
> - Comparison with [1]. We clarify our innovations compared to [1] in three aspects:
>    - **Novelty**. It needs to be recognized that hidden codes and codebook are inherently two concepts, the former referring to {$z_m$}$\_{m=1}^M$ and the latter to $\mathcal{A}=${$\mathbf{e}\_i$}$\_{i=1}^{|\mathcal{A}|}$ in this paper. Two kinds of masked modeling are adopted in [1]: (1) Masked Input Modeling(MIM) that masks input features {$\mathbf{x}_m$}$\_{m=1}^M$ and (2) Masked Hidden Modeling (MHM) that masks hidden codes {$z_m$}$\_{m=1}^M$. However, unlike previous works [1] that mask input features {$\mathbf{x}_m$}$\_{m=1}^M$  or hidden codes {$z_m$}$\_{m=1}^M$ for training a better encoder or decoder, we randomly mask the microenvironment codes {$\mathbf{e}_i$}$\_{i=1}^{|\mathcal{A}|}$ in the codebook $\mathcal{A}$ to capture their dependencies for learning a better codebook. The above explanations have been described in **Lines 139-141**.
>    - **Motivation.** There are generally overlaps (dependencies) in the microenvironments of two residues that are adjacent in sequence or structure. To capture such microenvironmental dependencies in the codebook, we propose MCM to mask the codebook $\mathcal{A}$ and then reconstruct the inputs based on the unmasked codes, aiming to learn a more expressive codebook $\mathcal{A}$ rather than a more powerful protein encoder. The above explanations have been described in **Lines 218-222**.
>    - **Comparison.** We provide a comparison of MCM with two schemes (MIM and MHM) in **Lines 345-351** and **Table 3**, which demonstrates the superiority of MCM over MIM and MHM.
>
> - Comparison with [2]. [2] defines the sequence and structural context around a target residue as its microenvironment and classifies it into **pre-defined** categories based on its physicochemical properties (see Fig. 1(b) in [2]). The essential difference between our work and [2] is that [2] is a microenvironment classification task for known categories, whereas MAPE-PPI aims to discover and encode microenvironments with **unknown** categories. We have clearly stated the differences between our work and [2] in **Lines 58-68**, as follows
>
>    > “There has been some work that defines the sequence and structural contexts surrounding a target residue as its microenvironment and classifies it into a dozen types based on its physicochemical properties or geometric features [2], which constitutes a special microenvironment vocabulary. However, unlike the rich word vocabulary in languages, the experimental microenvironment vocabulary is usually coarse-grained and limited in size. Therefore, to learn a fine-grained vocabulary with rich information, we propose …”
>    >
>
> We have cited, discussed, and compared [1,2] in the revised manuscript. We sincerely hope that this helps to clarify your concerns on "limited innovation".
>
> ---
>
> [1] Peng Z, Dong L, et al. Beit v2: Masked image modeling with vector-quantized visual tokenizers. 2022.
>
> [2] Lu H, Diaz D J, et al. Machine learning-aided engineering of hydrolases for PET depolymerization. 2022.

---

> ### Author Response · Authors · 2023-11-12
> **Response to Reviewer wZyc - Part (2/2)**
>
> **Q3**: What is the investigation of vocabulary redundancy? The authors create a sufficiently large vocabulary. Are they redundant? Or what is the relationship between vocabulary size or redundancy and archivable performance?
>
> **A3**: We have studied how codebook size influences performance in **Lines 355-360** and **Table 4**, which reveals the effects of codebook redundancy. Did you miss this part of the experiment? As shown in **Table 4**, codebook size does influence model performance; a codebook size that is set too small, e.g., 64 or 128, cannot cover the diversity of microenvironments, resulting in poor performance. Although setting the codebook size $|\mathcal{A}|$ to 2048 can sometimes outperform 512 or 1024, the superiority is not significant, which we suspect is due to code redundancy in the codebook. In practice, the codebook size $|\mathcal{A}|$ is a hyperparameter that can be determined by selecting the model that performs the best on the validation set for testing.
>
> ---
>
> **Q4**: The efficiency claim is very confusing. Also, why the proposed design can enable efficiency? What are the computational and memory overheads of the sufficiently large vocabulary and two-stage training? More detailed time complexity analyses are needed.
>
> **A4**: The efficiency (or computational efficiency) discussed in this paper refers to the training efficiency rather than inference efficiency. As described in subsection 3.1 of the problem statement on **Lines 149-150**, this paper adopts a **transductive** setting (common for graphs), which requires that all test data (without labels) are involved in training. As a result, the main bottleneck here is training efficiency rather than inference efficiency, so in order to "scale to PPI prediction with millions of PPIs", it prefers a model with very high training efficiency. Next, we explain why MAPE-PPI improves efficiency from three aspects：
>
> - **Roles of the Codebook.** The microenvironments of different residues may also be similar or overlap. If the microenvironments of two residues are highly identical, it would be redundant to encode them separately at each training epoch. If we can learn a ``vocabulary" (i.e., codebook) that records those most common microenvironmental patterns, we can use it as an off-the-shelf tool to encode proteins into microenvironment-aware embeddings. Thus, training efficiency can be improved by decoupling end-to-end PPI prediction into two stages of learning the microenvironment codebook and codebook-based protein encoding. These explanations have been added in **Lines 52-58** of the revised manuscript.
>
> - **Training Time Complexity.** A comparison of the time complexity of HIGH-PPI with that of MAPE-PPI is provided in **Appendix E, Lines 576-586**. The key analysis is as follows
> > "The time complexity of MAPE-PPI is $\mathcal{O}\big(N\left(|\mathcal{V}_P|\cdot|\mathcal{A}|+|\mathcal{E}_P|\right)+|\mathcal{E}_T|\big)$. Besides, the time complexity of HIGH-PPI for protein encoding and PPI prediction is $\mathcal{O}\big(N\left(|\mathcal{V}_P|+|\mathcal{E}_P|\right)\big)$ (without vector quantization and protein decoding) and $\mathcal{O}(|\mathcal{E}_T|)$, respectively. However, since HIGH-PPI jointly performs protein encoding and PPI prediction in an end-to-end manner (while MAPE-PPI decouples the two), its time complexity is the product of the two, that is $\mathcal{O}\big(N\left(|\mathcal{V}_P|+|\mathcal{E}_P|\right)\cdot|\mathcal{E}_T|\big)$, which is much higher than MAPE-PPI."
> >
>
> - **Training Time Analysis.** Refer to **Figure 1** or **Figure A1**, MAPE-PPI trains most efficiently regardless of whether or not pre-training is done and what type of pre-training is used.
>
> ---
>
> **Q5**: Only a single level of sequence similarity is considered. More like 30%, 50%, 90% are encouraged. Multiple runs of experiments are encouraged to show the prediction stability.
>
> **A5**: We have added experiments on how sequence similarity influences performance. The detailed experimental setup, results, and analysis can be found in **Appendix H**, **Lines 594-604**. The issue of prediction stability has been taken into account in the original manuscript, as stated in **Lines 267-268**, that
>
> > "we run each set of experiments five times with different random seeds, and report the average performance.”
> >
>
> ---
>
> In light of these responses, we hope we have addressed your concerns. If we have left any notable points of concern unaddressed, please do share and we will attend to these points. We sincerely hope that you can appreciate our efforts on responses and revisions and thus raise your score.

---

> ### Author Response · Authors · 2023-11-21
> **Looking forward to your reply**
>
> Dear Reviewer,
>
> Thank you for your insightful and helpful comments once again. We greatly appreciate your feedback. We have carefully responded to each of your questions point-by-point.
>
> Unlike previous years, there will be no second stage of author-reviewer discussions this year. Considering that the deadline for discussion is approaching and that you are the only one of the three reviewers to hold a negative score, your opinion is very important to us. If you are satisfied with our response, please consider raising your score. If you need any clarification, please feel free to contact us.
>
> Thanks again for your valuable time spent on reviewing the manuscript.
>
> Best regards,
>
> Authors.

---

### Author Response · Authors · 2023-11-19
**Look forward to further feedback**

Dear Reviewers,

We would like to express our sincere gratitude for dedicating your time to reviewing our manuscript. Unlike previous years, there will be no second stage of author-reviewer discussions this year, and a recommendation needs to be provided by November 22, 2023. Considering that the date is approaching, we look forward to hearing from you! We have thoroughly considered all the feedback and provided a detailed response a few days ago. We hope that our response addresses your concerns to your satisfaction. If you still need any clarification or have any other questions, please do not hesitate to let us know.

Best regards,

Authors.

---

### Comment · Area_Chair_zuzf · 2023-11-22
**Reviewers - Pls provide your response to authors' rebuttal**

Dear All,

The authors have dedicated significant efforts to provide detailed rebuttal. I would appreciate it if you could share your feedback with them.

Thank you for your valuable contributions to ICLR!

Best regards,
Area Chair

---

### Meta-Review · Area_Chair_zuzf · 2023-12-07

**Metareview:**

(a) Scientific Claims & Findings:
The paper focuses on developing protein-protein interaction models. It introduces microenvironments of amino acid residues, proposing microenvironment-aware protein embeddings for PPI prediction. They encode microenvironments into discrete codes, enabling masked codebook modeling training. The experiments demonstrate scalability in PPI prediction with millions of PPIs. Reviewer wZyc raised concerns about the efficiency claim and pointed out missing important references. Reviewer 4o6m acknowledged the method's superiority but suggested comparisons with the latest structure-based methods, sensitivity analysis for hyperparameters, more dataset details, and real-world examples to enhance credibility.

(b) Strengths:
- Clear description of the fundamental algorithm and fluent writing style.
- Superiority of the proposed method compared to previous ones in PPI prediction.
- Comprehensive coverage of various PPI baselines, both sequence- and structure-based approaches.

(c) Weaknesses:
- Lack of comparison with the latest structure-based methods.
- Insufficient sensitivity analysis on hyperparameters.
- Missing detailed dataset descriptions and real-world examples.
- Unclear explanation of the efficiency claim raised concerns about its inference and training efficiency. Reviewer wZyc pointed out these issues, emphasizing missing references and the need for more comprehensive analyses.

The paper showcases a novel approach but needs to address concerns regarding method validation, comparison with recent methods, sensitivity analysis, and providing more comprehensive dataset details and real-world validations.

**Justification For Why Not Higher Score:**

N/A

**Justification For Why Not Lower Score:**

1. Acknowledged Significance and Contribution: The paper's novelty and contribution to the field of protein modeling have been recognized. It explores a less-explored area of protein discrete modeling, which holds potential for compressing or discretizing the sequence and structural space of proteins. This innovative direction in protein modeling is acknowledged as a significant contribution.

2. Positive Evaluation of Method Superiority: Reviewers acknowledge the paper's methodological superiority over existing approaches in protein-protein interaction prediction. The proposed method demonstrates effectiveness and superiority in comparison to prior methods, enhancing its importance and relevance within the field.

3. Soundness, Clarity, and Presentation: The paper received positive ratings for soundness, clarity in presenting the algorithm, and overall presentation. These aspects contribute to its strength, demonstrating a comprehensive understanding and effective communication of the proposed methodology.

4. Balancing Weaknesses with Strengths: While some weaknesses were identified, such as unclear efficiency claims, missing references, and scope for experiment enhancement, the positive evaluations regarding the paper's contribution, significance, and methodological superiority balance these criticisms. The identified weaknesses, while important for improvement, might not necessarily warrant a lower overall score due to the paper's recognized strengths.

The decision to not lower the score is primarily influenced by the positive recognition of the paper's significance, methodological superiority, and contribution to the field, despite the identified areas for improvement.

---

### Decision · Program_Chairs · 2024-01-16

Accept (spotlight)